# AAV-HDV: An Attractive Platform for the In Vivo Study of HDV Biology and the Mechanism of Disease Pathogenesis ^†^

**DOI:** 10.3390/v13050788

**Published:** 2021-04-28

**Authors:** Sheila Maestro, Nahia Gómez-Echarte, Gracián Camps, Carla Usai, Lester Suárez, África Vales, Cristina Olagüe, Rafael Aldabe, Gloria González-Aseguinolaza

**Affiliations:** 1Programa de Terapia Génica y Regulación de la Expresión Génica, Centro de Investigación Médica Aplicada (CIMA), Universidad de Navarra, Avenida Pío XII, 31080 Pamplona, Spain; smaestro@alumni.unav.es (S.M.); ngomez.9@alumni.unav.es (N.G.-E.); gcamps@alumni.unav.es (G.C.); c.usai@qmul.ac.uk (C.U.); avales@unav.es (Á.V.); colague@unav.es (C.O.); 2Instituto de Investigación Sanitaria de Navarra, IdiSNA, 31080 Pamplona, Spain; 3Suite 110 Research Triangle Park, 20 TW Alexander Drive, AskBio, NC 27709, USA; lsuarez@askbio.com

**Keywords:** HDV, mouse model, AAV, HDAg, liver damage

## Abstract

Hepatitis delta virus (HDV) infection causes the most severe form of viral hepatitis, but little is known about the molecular mechanisms involved. We have recently developed an HDV mouse model based on the delivery of HDV replication-competent genomes using adeno-associated vectors (AAV), which developed a liver pathology very similar to the human disease and allowed us to perform mechanistic studies. We have generated different AAV-HDV mutants to eliminate the expression of HDV antigens (HDAgs), and we have characterized them both in vitro and in vivo. We confirmed that S-HDAg is essential for HDV replication and cannot be replaced by L-HDAg or host cellular proteins, and that L-HDAg is essential to produce the HDV infectious particle and inhibits its replication. We have also found that lack of L-HDAg resulted in the increase of S-HDAg expression levels and the exacerbation of liver damage, which was associated with an increment in liver inflammation but did not require T cells. Interestingly, early expression of L-HDAg significantly ameliorated the liver damage induced by the mutant expressing only S-HDAg. In summary, the use of AAV-HDV represents a very attractive platform to interrogate in vivo the role of viral components in the HDV life cycle and to better understand the mechanism of HDV-induced liver pathology.

## 1. Introduction

Hepatitis delta virus (HDV) is a satellite virus that requires hepadnavirus envelope proteins for its transmission. Approximately 5% of Hepatitis B Virus (HBV) carriers have been exposed to HDV, with a total of 15–20 million patients worldwide, although recent studies have reported higher prevalence numbers [1,2,3]. HDV infection is associated with the most severe form of viral hepatitis, with a twofold higher risk of developing cirrhosis, a threefold higher risk of developing hepatocellular carcinoma (HCC), and twofold increased mortality in comparison with HBV monoinfection [4,5,6]. Despite recent advances in the management of this condition, it still represents a significant medical burden [6,7,8]. Conventional treatment for chronic HDV infection consists of long-term administration of standard or pegylated interferon alpha (peg-IFN-α), which leads to a sustained virological response (SVR) in only 20–30% of patients treated and is frequently associated with serious adverse reactions. Several new agents are now under clinical investigation: bulevirtide, lonafarnib and nucleic acid polymers [9,10]. These drugs target essential steps of the HDV viral cycle - like viral entry, the isoprenylation of the large viral antigen, and viral encapsidation. Importantly, bulevirtide was approved in Europe in July 2020 [11].

The mechanisms associated with the severity of the disease remains unknown, although it is thought to be associated with the host immune response since a significant inflammatory infiltrate composed of macrophages and lymphocytes is observed in the liver of HDV patients [12,13]. However, HDV has been also considered to be directly cytopathic, particularly during the acute stage of infection, and it has been related to the expression of HDV antigens (HDAgs) [14,15,16]. The cytotoxic properties of HDAgs are largely unsettled due to the different results obtained in vitro, which depend on the cellular systems employed [15,16,17] 

HDV is a satellite RNA virus that requires the surface antigens of HBV (HBsAg) for viral assembly and transmission [10]. Specific interaction between HBsAg and the human Na^+^/taurocholate co-transporting polypeptide (hNTCP) determines the hepatotropism and species-specificity of both viruses [18,19]. Although NTCP is also expressed in mouse livers, mice are resistant to HBV and HDV infections due to small differences between the human and mouse NTCP protein sequences [18]. 

The HDV genome is a circular negative-sense RNA molecule of approximately 1700 nucleotides that appears as a double-stranded rod-like structure. It contains a single open reading frame (ORF) that encodes two HDAgs—a 24 kDa small HDV antigen (S-HDAg), and a 27 kDa large HDV antigen (L-HDAg) of 195 and 214 amino acids, respectively [10]. Despite sharing most of their amino acid sequence, they differ significantly in their functions: it has been described that S-HDAg is essential for HDV replication, whilst L-HDAg blocks HDV replication and is essential for viral assembly [17,20,21]. The production of L-HDAg requires RNA editing at adenosine 1012 (amber/W site) in the antigenomic RNA sequence by host adenosine deaminase that act on RNA-1 (ADAR-1) [22].

All these studies have been performed in cell culture due to the lack of an in vivo model in which the HDV genome can be easily manipulated and tested [23]. Recently, we developed a mouse model of HDV replication, based on the adeno-associated viral vector (AAV)-mediated delivery of HBV and HDV replication-competent genomes to the liver. This model mimics most of the features of HDV infection in humans, including the induction of liver inflammation and liver injury in association with the expression of genes involved in the development of HCC, cirrhosis, fibrosis, and cell death [24]. Using this model, we described that mitochondrial antiviral signalling protein (MAVS) was responsible for the strong type I interferon (IFN) response triggered by HDV replication, observed in cell culture and animal models [23,24,25,26,27]. This result was corroborated in human hepatic cells identifying melanoma differentiation-associated protein 5 (MDA5) as the main cellular pattern recognition receptor (PRR) involved in HDV detection that signals through MAVS and induces IFN-β gene transcription [28]. Furthermore, the involvement of tumor necrosis factor-alpha (TNF-α) in the HDV-induced pathology was demonstrated using this animal model and it was confirmed by the amelioration of liver damage in AAV-HBV/HDV-injected animals treated with Etanercept, a drug that blocks TNF-α receptor interaction [29].

One of the beauties of this model is that it allows for genetic modification of HDV viral genomes that can be then tested in cell culture and more importantly in the liver of mice to interrogate the role of viral components in the virus life cycle and in the induction of liver pathology. Furthermore, the establishment of HDV replication in genetically manipulated mice allows us to analyze the involvement of different host factors.

The aim of this work was to determine the role of HDV antigens in HDV-mediated liver injury using AAV-HDV carrying different mutations. 

## 2. Materials and Methods 

### 2.1. Plasmids and AAV Vectors

The AAV-HDV wild-type (WT) plasmid was generated as previously described by cloning the HDV 1.2× (genotype 1) genome sequence obtained from the plasmid pDL456 (kindly provided by J. M. Taylor) under the control of the hepato-specific promoter—enhancer of albumin/alpha-1-antitrypsin promoter (EAlb/AAT)—into the AAV-MCS plasmid [24]. Different HDV mutants were created by introducing point mutations. All the plasmids were generated with the In-Fusion^®^ HD Cloning Kit (Takara Bio USA, #638910, Mountain View, CA, USA) following the manufacturer’s instructions. To that end, the HDV-ΔS-HDAg, HDV-ΔL-HDAg, and HDV-ΔHDAg mutants were produced to block the expression of either S-HDAg, L-HDAg, or both, respectively. The first mutant contains a single nucleotide change (A/G) at nucleotide 1015 in the editable amber stop codon (TAG), which produces a tryptophan codon (TGG) that cannot be edited, always leading to the production of L-HDAg. The second mutant (HDV-ΔL-HDAg) contains a point mutation (G/A) at nucleotide 1016 that changes the amber/W site (TAG) into another stop codon (TAA), which cannot be converted into a tryptophan codon by ADAR-1 [21]. Therefore, from this sequence, only S-HDAg will be produced. For the construction of the HDV-ΔHDAg mutant, the start codons found in the sequence (located between nucleotides 1088–1090 and 1364–1366) were replaced by stop codons (TAA). Finally, AAV-HBV plasmid was constructed by insertion of 1.3× copies of the HBV genome (genotype D, serotype ayw) obtained from the pSP65 plasmid (kindly provided by Dr. Francis Chisari) into pAAV-MCS [24].

The recombinant AAV genomes were encapsidated in the mouse liver tropic AAV serotype 8 capsid. Briefly, each AAV vector plasmid and the helper/packaging plasmid pDP8.ape (Plasmid factory, Bielefeld, Germany) were co-transfected into HEK-293T cells. The cells and supernatants were harvested 72 h after transfection, and the virus was released from the cells by three rounds of freeze-thawing. Crude lysate from all batches was then treated with DNAse and RNAse (0.1 mg per p150 culture dish) for 1 h at 37 °C and then kept at −80 °C until purification. Purification of crude lysate was performed by ultracentrifugation in optiprep density gradient medium-iodixanol (Sigma-Aldrich, St Louis, MO, USA). Thereafter, iodioxanol was removed, and the batches were concentrated by passage through Amicon Ultra-15 tubes (Ultracel-100K; Merck Millipore, Cork Ireland). For virus titration, viral DNA was isolated using the High Pure Viral Nucleic Acid kit (Roche Applied Science, Mannheim, Germany). Viral titers in terms of viral genome copies per millilitre (vg/mL) were determined by Real Time-qPCR (#1855196, BioRad Hercules, CA, USA).

### 2.2. Animals and Treatment

C57BL/6 mice were purchased from Harlan Laboratories (Barcelona, Spain). Recombination activating gene 1 (RAG-1) deficient mice on a C57BL/6 genetic background (RagB6) were bred and maintained at the animal facility of the University of Navarra. Six- to eight-week-old male mice were used in all experiments. Mice were kept under controlled temperature, light, and pathogen-free conditions. Mice were injected intravenously (i.v.) with AAV vectors (5 × 10^10^ vg of each vector per mouse) diluted in saline solution in a volume of 100 µL. For all procedures, animals were anesthetized by intraperitoneal (i.p.) injection of a mixture of xylazine (Rompun 2%, Bayer) and ketamine (Imalgene 50, Merial) 1:9 *v*/*v* or by isoflurane inhalation. Blood collection was performed by submandibular bleeding, and serum samples were obtained after centrifugation of total blood. Animals were euthanized by cervical dislocation after being anesthetized. The experimental design was approved by the Ethics Committee for Animal Testing of the University of Navarra (R-132-19GN). 

### 2.3. Cell Lines

Two human hepatoma cell lines, HepG2 (ATCC^®^ HB-8065™) and Huh-7 (ATCC^®^ PTA-4583), were transfected with HDV WT-encoding plasmids, while the characterization of HDV-ΔL-HDAg, HDV-ΔS-HDAg, and HDV-ΔHDAg mutants was only performed in Huh-7 cells. Moreover, Huh-7-hNTCP cells, kindly provided by Dr. Urtzi Garaigorta, were employed for infectivity studies. HepG2 and Huh-7 cell lines were cultured in Dulbecco’s modified Eagle’s medium (DMEM) supplemented with 10% fetal bovine serum (FBS), 1% of L-glutamine, 1% of glucose, 100 U/mL of penicillin-streptomycin, and non-essential amino acids, and incubated at 37 °C with 5% CO_2_ in a humidified atmosphere. In the case of Huh-7-hNTCP cells, the culture medium was supplemented with 2.5 µg/mL blasticidine to ensure the selection of hNTCP-expressing cells. HEK293T cells (ATCC^®^ PTA-4583) were used for AAV production.

### 2.4. DNA Transfection

For the comparative study of HDV transfection in HepG2 and Huh-7 cells, both cell lines were seeded and transfected in parallel. Briefly, 3.5 × 10^5^ HepG2 and Huh-7 cells were seeded per well in 6-well plates and maintained in DMEM 10% FBS. After 24h, the medium was substituted by Opti-mem, and cells were transfected with 2.5 µg DNA/well using Lipofectamine 3000, following the manufacturer’s instructions. 4–6 h later, cells were supplemented with 1.5 mL of DMEM 10% FBS, and the day after, the medium containing the transfection reagents was replaced by fresh DMEM 10% FBS. Cells were then collected at different time points until 14 days post-transfection (dpt). At 7-dpt, cells were trypsinized, split 1:3 and reseeded in new wells. 

### 2.5. Cell Fractionation, Protein Extraction and Quantification

Harvested cells were resuspended in the RIPA lysis buffer supplemented with 1mM PMSF (phenylmethylsulfonyl fluoride), 1% *v/v* of a pre-formed protease inhibitor cocktail (#P8340, Sigma-Aldrich: 4-(2-Aminoethyl)-benzenesulfonyl fluoride hydrochloride (AEBSF) at 104 mM, Aprotinin at 80 μM, Bestatin at 4 mM, E-64 at 1.4 mM, Leupeptin at 2 mM and Pepstatin A at 1.5 mM), and 0.1 mM of sodium orthovanadate, and incubated on ice for 30 min. Cell extracts were then spun for 20 min at 13,000 rpm and 4 °C, and supernatants were collected for western blot analysis. Protein concentration was determined by BCA assay (#23227, Thermo Fisher Scientific, Waltham, MA, USA).

To determine the intracellular location of the proteins, cells were harvested with Trypsin-EDTA and centrifuged for 5 min at 500 g and 4 °C. Then, proteins of each cellular compartment were separated with the Subcellular Protein Fractionation Kit (#78840, Thermo Fisher Scientific), following the manufacturer’s instructions. 

### 2.6. Western Blot 

Equal amounts of protein were loaded in a SDS-polyacrylamide gel, separated by electrophoresis, and transferred onto a nitrocellulose membrane by electroelution. After a 45 min blockade with TBS-Tween™ 20 with 5% non-fat dry milk at room temperature (RT), the membranes were incubated overnight at 4 °C with the serum from patient CUN-28336 at dilution 1:2500. Patient serum was provided by the Biobank of the University of Navarra and processed following standard operating procedures approved by the Ethical and Scientific Committee (2019.217 CEI-CUN). After an extensive wash with TBS-Tween™ 20, the membranes were incubated with horseradish peroxidase (HRP)-conjugated secondary antibody at RT for 1h. The substrate of the enzyme was provided by the reagents of SuperSignal™ West Femto (#34095, Thermo Fisher Scientific), and the signal was detected by the ODYSSEY CLx near-infrared fluorescence imaging system. 

### 2.7. RNA Extraction and RT-qPCR

Total RNA from liver samples was isolated using TRI Reagent^®^ (#T9424, Sigma-Aldrich) according to the manufacturer’s instructions. Total RNA was pre-treated with DNAse I (#AM-1907, TURBO DNA-free™ Kit, Applied Biosystems, Foster City, CA, USA) and reverse-transcribed into complementary DNA (cDNA) using M-MLV reverse-transcriptase (Invitrogen, #28025013). Real-time quantitative polymerase chain reactions (RT-qPCR) were performed using iQ SYBR Green Supermix (#170-8884, BioRad) in a CFX96 Real-Time Detection System (#1855196, BioRad) and primers as specified in Appendix A. HDV strand-specificity was analyzed as described elsewhere [24]. GAPDH was used as a control housekeeping gene.

### 2.8. Immunofluorescence (IF)

Cells were fixed in PBS 4% paraformaldehyde (PFA) for 20 min at RT and permeabilized with PBS 0.1% Triton X-100 for 15 min at RT prior incubation with PBS-0.1% tween-20 5% BSA (blocking buffer) for 30 min at 37 °C. After washing, cells were incubated for 30 min at 37 °C with the serum from patient CUN-28336 at dilution 1:2500. Then, cells were incubated with the secondary antibody (goat anti-human IgG conjugated to Alexa Fluor 488, Applied Biosystems-Life Technologies, Waltham, MA, USA) at dilution 1:3000 for 30 min at 37 °C, protected from light. Finally, samples were covered with a mounting medium containing DAPI (1.5 µg/mL, Vector Laboratories), and the fluorescent samples were visualized with a confocal microscope (Zeiss LSM 880, Carl Zeiss, Oberkochen, Germany).

### 2.9. Histology and Immunohistochemistry (IHC)

Hematoxylin & Eosin (H&E): Liver sections were fixed with 4% paraformaldehyde (PFA), embedded in paraffin, sectioned (3 μm), and stained with hematoxylin and eosin. Sections were mounted and analysed by light microscopy for histologic evaluation.

Immunohistochemistry (IHC): the first steps were the same as for the H&E staining. Then, a step of antigen retrieval was performed that consisted of incubation for 30 min at 95 °C in 0.01 M Tris-1 mM EDTA pH 9. Subsequently, primary antibodies were incubated overnight at 4 °C. After rinsing in TBS-T, the sections were incubated with the corresponding secondary antibodies for 30 min at RT. Peroxidase activity was revealed using DAB+ and sections were lightly counterstained with Harris hematoxylin. Finally, slides were dehydrated in graded series of ethanol, cleared in xylene and mounted with Eukitt (Labolan, #28500, Navarra, Spain). Image acquisition was performed on an Aperio CS2 slide scanner using ScanScope Software (Leica Biosystems, Vista, CA, USA). The image analysis was performed using a plugin developed for Fiji, ImageJ (NIH, Bethesda, MD, USA). The antibodies employed are summarized in Table 1.

### 2.10. Infectivity Studies

Huh-7-hNTCP cells were seeded in 12-well plates with coverslips and maintained in DMEM 10% FBS 2% DMSO. The day of infection, culture medium was replaced by 1 mL of supernatant collected from Huh-7 co-transfected with HBV/HDV WT or HBV/HDV-mutants at 7 and 14 dpt. The day after infection, the medium was replaced by fresh DMEM 2% FBS 2% DMSO, and at 7 dpi, cells were fixed in 4% PFA, and the presence of HDAg-positive cells was examined by performing IF, as indicated above.

### 2.11. Statistical Analysis 

Statistical analyses were performed using GraphPad Prism 8.0 software. The data are presented as individual values ± standard deviation. Statistical significance was determined using an unpaired t-test for single comparisons and two-way ANOVA followed by Bonferroni’s multiple comparison tests to find differences between groups. * *p* < 0.05, ** *p* < 0.01, *** *p* < 0.001, **** *p* < 0.0001. 

## 3. Results

### 3.1. Selection of the Human Hepatic Cell Line for the In Vitro Analysis of HDV Mutants

HepG2 and Huh-7 cells were transfected with equal amounts of a plasmid carrying 1.2× copies of the replication-competent HDV WT genome under the transcriptional control of a liver-specific promoter [24]. Cells were collected 1-, 3-, 7-, 10-, and 14-dpt and were only split at day 7 (Figure 1A). The presence of HDV genomes and antigenomes, the induction of IFN-β and MxA expression, and HDAgs expression were analyzed.

Not surprisingly, HDV antigenomes were detected 24 h after plasmid transfection in both cell lines, since the antigenome is the transcriptional product of the shuttle plasmid. Then, the levels increased up to 7 dpt in HepG2 cells, decreasing thereafter, while in Huh-7 cells the HDV antigenome copies continued increasing up to 14-dpt (Figure 1B). Moreover, HDV genomes were detected in both cell lines, indicating that HDV replication is initiated after plasmid transfection. In HepG2 cells, HDV genomes were detected at day 7 and, as observed for the antigenome, decreased thereafter. In Huh-7 cells, HDV genomes were detected 1-dpt, and the levels increased for the entire duration of the experiment. In both cell lines, the shuttle plasmid was lost with time, indicating that HDV replication is self-sustained and not entirely dependent on the presence of the transfected plasmid (Figure A1, in Appendix A). Regarding HDAgs expression, S-HDAg was detected in both cell lines at day 3 (Figure 1C). In HepG2, S-HDAg is the only isoform detected, and its expression decreased at day 14 (Figure 1C). In contrast, in Huh-7 cells, S-HDAg expression increased with time, and L-HDAg detected 10-dpt increasing to 14-dpt (Figure 1C). Thus, while Huh-7 cells support HDV replication and HDAg expression for at least 14 days, viral replication in HepG2 cells decreases over time, and gene editing either does not occur at all or it does at a very low frequency, since no L-HDAg was detected.

In order to understand the different behaviour of these two types of human hepatic cell lines, and since it has been recently reported that type I IFN strongly suppresses the cell-division-mediated spread of HDV genomes [29], we analyzed IFN-β and MxA expression in both cell lines after HDV-plasmid transfection. While in Huh-7 cells, IFN-β mRNA was barely detected, and MxA mRNA was undetectable, in HepG2 cells, IFN-β and MxA were highly expressed at 10-dpt, 3 days after the cells were split (Figure 1D). These results indicate that, most likely, the differences found between HepG2 and Huh-7 cells were due to HDV-sensing by the innate immune system and the induction of type I IFN response, which blocks the spread of HDV in dividing HepG2 cells. Thus, the Huh-7 cell line was selected as the cellular platform for the characterization of HDV mutants.

### 3.2. In Vitro Analysis of HDV Mutants

Three different HDV mutants were generated: (1) HDV defective in the expression of both antigens (HDV-∆HDAg), (2) HDV defective in the expression of S-HDAg (HDV-∆S-HDAg), and (3) HDV defective in the expression of L-HDAg (HDV-∆L-HDAg). 

The experiment was performed as described in Figure 1A. As expected, no HDAg expression was detected in HDV-∆HDAg transfected cells (Figure 2A), and the HDV antigenome sequence was detected at very low levels. In cells transfected with HDV-∆S-HDAg, L-HDAg was transiently expressed at day 3 and only the HDV antigenome sequence was detected due to the transcriptional activity of the EAlb/AAT promoter, however, it dropped with time due to cell division (Figure 2B). Furthermore, transfection with the HDV-∆L-HDAg mutant resulted in the expression of S-HDAg from 7-dpt, which increased with time, and both HDV genome and antigenome were detected at similar levels to those found in cells transfected with HDV WT. The analysis of HDAg expression by IF in HDV WT and HDV-∆L-HDAg-transfected cells showed a preferential localization of HDAgs in the nuclear compartment at 7- and 14-dpt. However, while HDAgs were detected in both the nucleus and in the cytoplasm in HDV WT-transfected cells, HDAg remained mainly in the nucleus in cells transfected with the HDV-ΔL-HDAg mutant (Figure 2C). Cell fractionation studies corroborated this observation, showing the retention of HDAg in the nuclear fraction (Figure 2D, Figure A2). 

Then, we tested whether HDV infectious particles could be produced in the absence of L-HDAg. For that purpose, Huh-7 cells were co-transfected with the HBV-genome-containing plasmid and either HDV WT or HDV-∆L-HDAg plasmids. Then, supernatants were harvested at 7- and 14-dpt and added to Huh-7-hNTCP cells. As shown in Figure 2E, Huh-7 cells co-transfected with the HDV WT plasmid are able to produce HDV infectious particles that were more abundant at 14-dpt than at 7-dpt in association with L-HDAg expression at later time points; on the contrary, no HDV infective particles were produced in cells transfected with the HDV mutant lacking L-HDAg. 

### 3.3. In Vivo Virological Analysis of HDV Mutants

For in vivo analysis, hepatotropic AAV vectors (serotype 8) were generated to convey HDV mutant genomes to the liver of mice. AAV-HDV WT, AAV-HDV-∆HDAg, AAV-HDV-∆L-HDAg and AAV-HDV-∆S-HDAg vectors were co-injected together with AAV-HBV in C57BL/6 mice, and animals were sacrificed 21 days after AAV injection for liver collection to analyze HDV replication and HDAg expression. No HDAg expression or HDV genomes were detected in the liver of mice that received HDV-∆S-HDAg or HDV-∆HDAg (Figure 3A,B), indicating the absence of HDV replication in those animals. In contrast, the levels of HDV genome and antigenome in HDV-∆L-HDAg-injected mice were similar to those obtained in the HDV WT group (Figure 3A). Moreover, only S-HDAg was detected by WB and at significantly higher levels than in the animals injected with AAV-HDV WT (Figure 3B,C). Furthermore, immunohistochemistry analysis of HDAg expression in liver sections revealed a slightly higher percentage of HDAg positive cells in HDV-∆L-HDAg than in HDV-WT-injected animals (Figure 3D,E), which cannot be attributed to the presence of more AAV genomes since the levels were similar in both groups (Figure A3).

### 3.4. Effect of HDV Mutants on Liver Damage 

After AAV intravenous administration, animals were weekly bled to biochemically analyze the presence of liver damage, and they were sacrificed twenty-one days post-infection (dpi) to evaluate liver histology. The results revealed that the administration of HDV-∆L-HDAg induced a significantly higher transaminase elevation than the one induced by HDV-WT (Figure 4A,B). Liver histology in both groups was characterized by the presence of inflammatory foci and hepatocyte hypertrophy and necrosis, which were more pronounced in the HDV-∆L-HDAg group. Most of the hepatocytes showed pyknotic and enlarged nuclei (Figure 4B). The HDV-∆L-HDAg group showed a significant increase in both the hepatocyte nuclear size (Figure 4C) and the number of hepatocytes undergoing apoptosis (activated caspase 3, a-Casp3, positive cells) (Figure 4D), in comparison with the HDV WT group.

### 3.5. Analysis of Liver Inflammatory Infiltrate and Cytokine Expression

For a better characterization of the HDV-∆L-HDAg mutant, animals were treated as previously described and sacrificed at 7, 14, and 21 dpi, and livers were extracted. In this case, we found a significant increase in HDV RNA at 14 dpi in the group of animals that received HDV-ΔL-HDAg, indicating a more active replication in the absence of L-HDAg (Figure A3).

Then, the presence of macrophages (F4/80+ cells) and both CD8+ and CD4+ T cells were examined by the immunostaining of liver sections and quantified. The analysis revealed higher amounts of immune cells in mice injected with the HDV-∆L-HDAg mutant than in HDV-WT mice (Figure 5A). 

Next, we analyzed the hepatic expression of the following cytokines: IFN-β, IL-1β, IFN-γ, IL-6, TNF-α, and TGF-β. The administration of the AAV-HDV-∆L-HDAg vector led to a higher induction of IFN-β compared with HDV WT at day 14, in association with the presence of more HDV genomes at this time point. However, IFN-β levels declined from day 14, and no difference was found at day 21 between both groups (Figure 5B). Moreover, the expression of the cytokines IL-1β, IFN-γ, IL-6, TNF-α, and TGF-β was higher in HDV-∆L-HDAg mice than in HDV-WT at days 14 and 21 post-infection, in agreement with the increased liver damage observed at these time points; however, only the expression of TNF-α was significantly higher (Figure 5B).

### 3.6. Production of Infectious Viral Particles

Since we have previously shown that immunocompetent C57BL/6 mice receiving AAV-HBV/HDV produced antibodies against HBV surface antigens that block HDV infection [24], to analyze the production of infectious viral particles, RagB6 mice (B and T cell deficient) were used. Animals were injected as described before, and Huh-7.5.1-hNTCP cells were infected with serum obtained at day 21. As shown in Figure 6, while a significant number of HDAg-positive cells were detected amongst the cells incubated with serum from HBV/HDV WT-injected mice, no infective virions were detected in the serum obtained from HBV/HDV-∆L-HDAg-injected mice. Interestingly, when we analyzed the severity of liver damage in these immunodeficient animals, no differences in its magnitude were observed in comparison to WT animals. No significant differences were observed in serum transaminase values (Figure 7A) or liver histology (Figure 7B), and the number of activated caspase 3 positive cells (Figure 7C) was similar, indicating that the stronger liver damage observed in HDV-∆L-HDAg is not directly associated to a T cell-mediated response.

### 3.7. Effect of L-HDAg Expression over the Hepatic Damage Induced by HDV-∆L-HDAg

To clarify if L-HDAg can modulate the damage induced by the HDV mutant expressing only S-HDAg, mice were co-injected with equal amounts of AAV-HDV-∆L-HDAg and AAV-HDV-∆S-HDAg, in combination with the AAV-HBV vector. The AAV-HDV-∆HDAg vector was also co-administered to normalize the amount of AAV genomes that each animal received. Mice were divided in 5 groups (*n* = 4 mice/group). Group 1: AAV-HBV + AAV-HDV WT + HDV-∆HDAg vector, group 2: AAV-HBV + HDV-∆L-HDAg + HDV-∆HDAg, group 3: AAV-HBV + HDV-∆L-HDAg + HDV-∆S-HDAg, group 4: AAV-HBV + HDV-∆HDAg + HDV-∆S-HDAg, group 5: 3 × AAV-HDV-∆HDAg. 

Serum transaminase levels analyzed weekly after vector injection revealed a significant reduction of liver damage in animals receiving both HDV-∆L-HDAg and HDV-∆S-HDAg vectors in comparison with animals receiving HDV-∆L-HDAg alone (Figure 8A). Moreover, liver histology reflected the biochemical findings, with nearly normal histology in the animals receiving both mutants in comparison with those receiving only HDV-∆L-HDAg, indicating all together that an early expression of L-HDAg ameliorates S-HDAg-induced liver damage. Furthermore, the administration of the HDV-ΔS-HDAg vector together with HDV-ΔL-HDAg resulted in lower levels of HDV RNA, in comparison with the levels in mice receiving the HDV-ΔL-HDAg mutant alone or the HDV WT, showing that L-HDAg had a strong inhibitory effect over HDV replication when it was expressed at the beginning of the HDV life cycle (Figure 8C). 

## 4. Discussion

HDV infection is responsible for the most severe form of viral hepatitis in humans, leading to cirrhosis in 80% of cases within 10 years, and with a significant proportion of patients dying of hepatic decompensation [1,3]. The mechanism involved in the severity of the disease remains obscure, largely because of the lack of appropriated small animal models in which in vivo molecular analysis can be performed. Relevant animal models of HDV infection like chimpanzees or woodchucks developed liver damage that can even be fatal, but the performance of mechanistic studies in these animal models is cumbersome [30,31]. Recently, we have developed a HDV mouse model that overcomes species-related limitations based on the transfer of the HDV replication-competent genome to mouse hepatocytes via an AAV vector [24]. For the first time, HDV-mediated liver injury was observed in mice, in association with a strong inflammatory infiltrate and hepatocyte apoptosis. Furthermore, we found that TNF-α was partially responsible for the observed liver damage. The beauty of this model is that animals with different genetic backgrounds can be used, and that mutations affecting specific viral components can be easily introduced in the HDV genome and tested in vivo [24,29].

Pioneering work performed in the ’80–’90s by several prominent virologists demonstrated the role of HDAgs in the viral cycle and the induction of cell damage [14,15,20,21,32,33,34]. Studies performed using different cellular systems and transfection of plasmids carrying replication competent HDV genomes or expressing the HDAgs demonstrated the differential roles of the small and the large antigens on HDV biology. They showed that while S-HDAg is necessary for HDV genome replication, L-HDAg has the capacity to inhibit it [32,33,34]. Furthermore, they also showed that the expression of S-HDAg might exert a cytotoxic effect and that L-HDAg interferes with multiple cellular signalling pathways and alters cellular homeostasis by inducing oxidative stress or increasing susceptibility to the effect of inflammatory cytokines [34,35,36,37,38]. Here, using the AAV vector as an HDV delivery platform, we have tested in vivo the role of HDAg in the viral life cycle and in the induction of liver damage in mice.

We have generated three different mutants with altered expression of HDAgs: one lacking HDAg expression, the second expressing only L-HDAg, and the third expressing only S-HDAg. The initial characterization of the mutants was performed in vitro by plasmid transfection into two hepatic cell lines of human origin. Firstly, HepG2 and Huh-7 cells were transfected with plasmids carrying the HDV WT sequence. We observed that in Huh-7 cells, both viral replication and HDAgs expression were sustained, despite cell passages and loss of the shuttle plasmid; however, in HepG2 cells, HDV replication and antigen expression decreased after splitting the cells. Furthermore, while both HDV antigens were expressed in Huh-7 cells, only S-HDAg was detected in HepG2. The absence of L-HDAg in HepG2 cells cannot be explained by the lack of ADAR-1, since both cell lines expressed similar levels of the editing protein (data not shown). One of the main differences we observed between the two cell lines is the activation of the cellular innate immune response to HDV replication. In HepG2 cells, we observed a strong activation of type I IFN response that was almost absent in Huh-7 cells. Interestingly, expression of IFN-β and MxA was detected only at day 10 after transfection, 3 days after splitting the cells. A potential explanation for this sharp peak is that at day 10, although undetectable by western blot, L-HDAg is present and HDV ribonucleoproteins (HDV RNPs) translocate from the nucleus to the cytoplasm where the viral RNA is sensed by MDA5 and initiates the activation of the innate immune response. Alternatively, the increase of the cellular division rate due to cell splitting might expose the RNPs to cytosolic RNA sensors. 

Recent work by Zhang et al. has similarly shown that HDV-induced IFN response suppresses cell division mediated-HDV spread in HepaRG cells [39], and this could explain why HDV replication is sustained in Huh-7 but not in HepG2 cells in our experimental setting.

Both in vitro and in vivo studies showed that the HDV genome itself is unable to initiate replication and requires the expression of viral antigens, indicating that the interaction between the cellular polymerase and the viral genome cannot be replaced by a host protein. More interestingly, S-HDAg cannot be substituted by the large antigen despite both sequences being identical except for the 19 extra amino acids in the carboxy-terminal region of L-HDAg. On the other hand, we also confirmed in vitro that L-HDAg is essential for the export of the HDV RNP from the nucleus to the cytoplasm and for the formation of HDV infective particles, both in vitro and in vivo [34,40], and that these functions cannot be exerted by S-HDAg. 

Interestingly, S-HDAg expression levels were significantly higher in the mutant lacking L-HDAg expression, and while viral replication was similar at 21 dpi, it was significantly higher at 14 dpi (Figure A3), pointing towards the regulatory role of this isoform in the inhibition of HDV replication as previously reported in vitro [21,34,41]. The decrease in HDV replication from day 14 to day 21 observed in mice receiving the HDV-ΔL-HDAg mutant could be associated with higher levels of liver damage and apoptotic hepatocytes.

Furthermore, we found that production of the antigenome sequence alone or in the presence of L-HDAg had no effect on liver damage, while the expression of S-HDAg in the absence of L-HDAg resulted in a significant increase in liver damage in comparison with the damage induced by the injection of the AAV carrying the HDV WT genome. As shown, mice receiving the HDV-ΔL-HDAg mutant presented a higher number of aberrant hepatocytes with bigger nuclei as well as a higher frequency of apoptotic cells in comparison to HDV WT. Moreover, we also found that the liver of these animals showed a more prominent inflammatory infiltrate (with significantly higher numbers of macrophages and T cells), resulting in higher expression levels of TNF-α, which we previously showed to play a significant role in HDV-induced liver damage [29]. Interestingly, the experiment performed in RagB6 mice, apart from demonstrating the essential role of L-HDAg in the formation of HDV infectious particles, also showed that T cells are not directly involved in the exacerbation of the liver disease induced by the HDV mutant expressing only S-HDAg, suggesting a direct cytotoxic effect associated with the overexpression and nuclear accumulation of S-HDAg. These results are in line with previous data demonstrating a direct cytotoxicity of S-HDAg [13,14,15]. Cole et al. observed that cells stably transfected with a replication-competent HDV construct showed spontaneous death that diminished when the expression of L-HDAg was detected [15]. Furthermore, it was reported that the L-HDAg/S-HDAg ratio is higher in patients with chronic HDV infection and low transaminase levels than in patients with HDV acute infection and high transaminase levels [16], suggesting, as we observed here, a major role of S-HDAg in the development of HDV-induced liver damage that seems to be attenuated by L-HDAg expression. In fact, we found that the coadministration of the mutant expressing only L-HDAg and the mutant expressing only S-HDAg, which resulted in an early expression of both antigens after AAV administration, significantly ameliorated the liver damage induced by the mutant that only expresses S-HDAg. We also observed, as expected, that L-HDAg reduced both HDV replication and S-HDAg expression levels. With these experiments, we cannot discard that L-HDAg plays a role in HDV-induced pathology. Several studies have shown that the expression of L-HDAg may alter cellular homeostasis and induce liver damage. However, using the AAV-HDV delivery system, it is impossible to isolate the effect of L-HDAg from that of S-HDAg and HDV replication since HDV-ΔS-HDAg transiently expresses L-HDAg, and HDV replication is not initiated. 

Additionally, our data indicate that hepatocyte infection with HDV viral particles carrying an edited genome will result in a sterile/unproductive infection since HDV replication cannot be initiated, and it will be productive only if the same cell is coinfected by an HDV carrying a non-edited genome. This is a very interesting finding considering that in patients, both non-edited and edited forms of the HDV genomes can be found in the circulation [42,43]. The role of the production of HDV particles containing an edited genome that will lead to a sterile/unproductive infection is not well understood. However, we might hypothesize that as the infection progresses, the viral particles harbouring edited genomes will enter cells in which HDV is actively replicating and will reduce viral replication and spread through the liver, but it will also ameliorate the activation of the HDV-induced inflammatory response and S-HDAg cytotoxicity. Using this strategy, the virus would avoid the death of infected cells and clearance by the immune system, improving the survival of the virus as well as the host cell. These findings are consistent with the hypothesis that L-HDAg may promote viral persistence by ameliorating liver damage [15].

In summary, the use of AAV-HBV/AAV-HDV has allowed us to confirm in vivo the properties attributed to the HDAgs in the HDV life cycle previously described almost exclusively in cell culture. However, more importantly, we demonstrated the major role of HDAgs, mainly S-HDAg, in the HDV-induced liver pathology that was described in vitro in the early 90′s.

Additional experimentation is required to determine the reasons leading to HDV-associated liver damage observed for the first time in this mouse model, as opposed to others. Both the high efficacy of AAVs in delivering their genetic cargo into the hepatocytes and the strong liver-specific promoter controlling the synthesis of the initial copies of the antigenome are plausible explanations. 

It is nonetheless clear that in this model, HDV replication and/or S-HDAg expression are required for the development of the observed liver pathology, since AAV vectors carrying either HBV replication-competent genomes or replication-deficient HDV genomes failed to induce liver damage.

Thus, AAV-mediated delivery of hepatitis virus replication-competent genomes represents a very attractive platform to determine the mechanism of HDV-induced liver pathology and for the development of new and more efficient treatments.

## Figures and Tables

**Figure 1 viruses-13-00788-f001:**
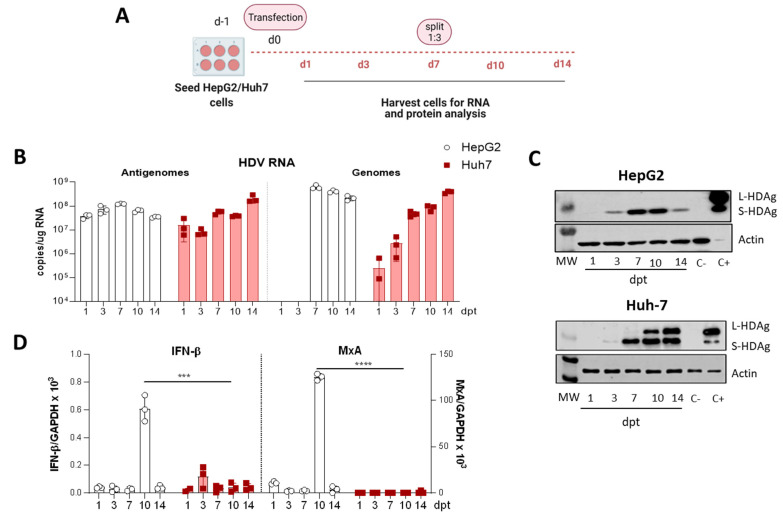
The HDV life cycle is supported in Huh-7 cells upon transfecting HDV-encoding plasmid but not in HepG2 cells. (**A**) Schematic representation of the experimental layout. HepG2 or Huh-7 cells were seeded and transfected with equal amounts of the plasmid encoding the HDV antigenome. For RNA and protein analysis, cells were collected at 1-, 3-, 7-, 10- and 14-days post-transfection (dpt), and cells were split 1:3 at 7-dpt. (**B**) Total RNA was extracted from cells and HDV antigenome and genome levels were assessed by RT-qPCR. (**C**) Western Blot analysis of HepG2 and Huh-7 cell lysates was performed to detect S-HDAg and L-HDAg. Positive control (C+): Huh-7 cells transfected with plasmids expressing S-HDAg and L-HDAg antigens and collected at 3 dpt. Negative control (C-): non transfected Huh-7 or HepG2 cells. (**D**) IFN-β (left) and MxA (right) expression levels were quantified by RT-qPCR and normalized using GAPDH as housekeeping gene. Statistical analysis using Mann–Whitney test revealed differences between the two cell lines (*** *p* < 0.001, **** *p* < 0.0001).

**Figure 2 viruses-13-00788-f002:**
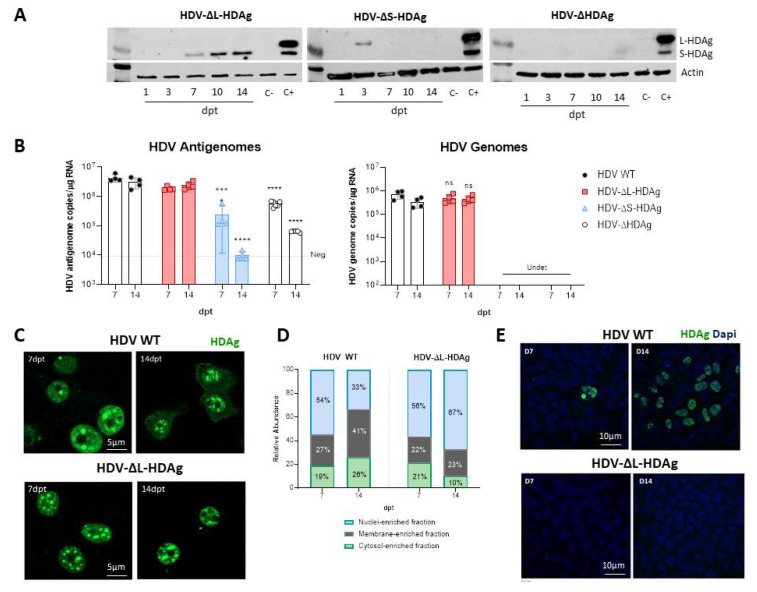
HDV mutants affecting S-HDAg and L-HDAg expression were generated and characterized in Huh-7 cells. (**A**) HDAg expression was assessed in Huh-7 cell lysates by Western Blot at 1-, 3-, 7-, 10- and 14-dpt. (**B**) HDV genomes were produced after transfection with the plasmids that express S-HDAg: HDV WT and HDV-ΔL-HDAg. (ns: no significant, *** *p* < 0.001, **** *p* < 0.0001). (**C**,**D**) HDAg localization was examined by immunofluorescence (scale bar: 5 μm) and by cell fractionation. The percentage of HDAgs present in the cytosolic-, the membrane- and the nuclear-enriched fractions was determined at 7- and 14-dpt. (**E**) Huh-7.5.1-hNTCP cells were infected with supernatants from HBV/HDV WT, and HBV/HDV-ΔL-HDAg co-transfected cells collected at 7- and 14-dpt. At 7 days post-infection (dpi), cells were fixed and immune stained with human serum for HDAg detection. Scale bar: 10 μm.

**Figure 3 viruses-13-00788-f003:**
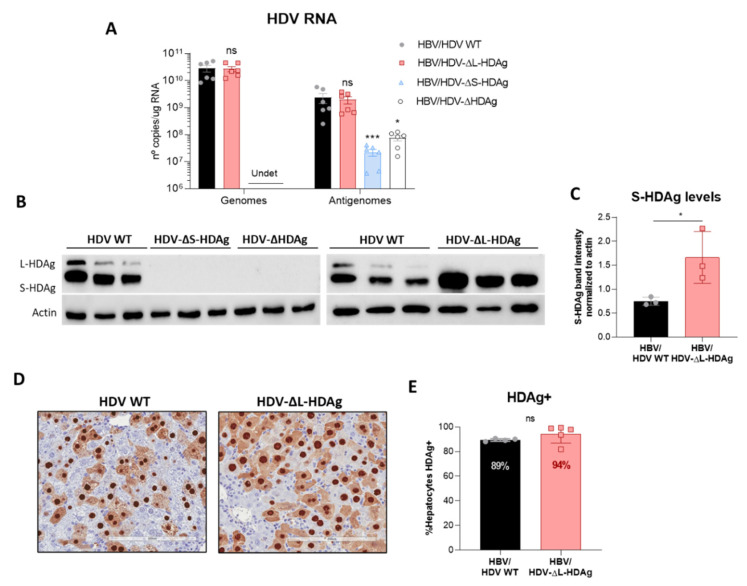
Administration of AAV-HDV mutants lacking the expression of S-HDAg resulted in a loss of HDV replication, while the absence of L-HDAg enhances S-HDAg production. Eight-week-old C57BL/6 WT mice received (*n* = 6 mice/group) 5 × 10^10^ genome copies of AAV-HDV, AAV-HDV-ΔL-HDAg, AAV-HDV-ΔS-HDAg and AAV-HDV-ΔHDAg in combination with 5 × 10^10^ genome copies of AAV-HBV. (**A**) HDV genomes and antigenomes were analyzed in the livers of infected mice at 21 dpi (significant differences were determined by Kruskal-Wallis test and Dunn’s post-test). (*Undet*: undetectable; ns: no significant, * *p* < 0.05, *** *p* < 0.001). (**B**) Western blot analysis of liver samples revealed the absence of HDAgs in mice receiving AAV-HDV-ΔS-HDAg and AAV-HDV-ΔHDAg, and only S-HDAg was detected in the livers of mice receiving the HDV-ΔL-HDAg vector. (**C**) The levels of S-HDAg were compared between the HDV WT and HDV-ΔL-HDAg groups by densitometry (* *p* < 0.05). (**D**) Immunostaining against HDAgs was performed at 21 dpi in the liver sections of mice injected with the AAV-HDV WT and AAV-HDV-ΔL-HDAg vectors. Scale bar: 200 μm. (**E**) HDAgs-positive cells were quantified in both groups of mice (*n* = 4–5). Mann–Whitney test revealed significant differences in S-HDAg protein levels and no differences between the percentage of HDAg-positive cells at 21 dpi (ns: no significant).

**Figure 4 viruses-13-00788-f004:**
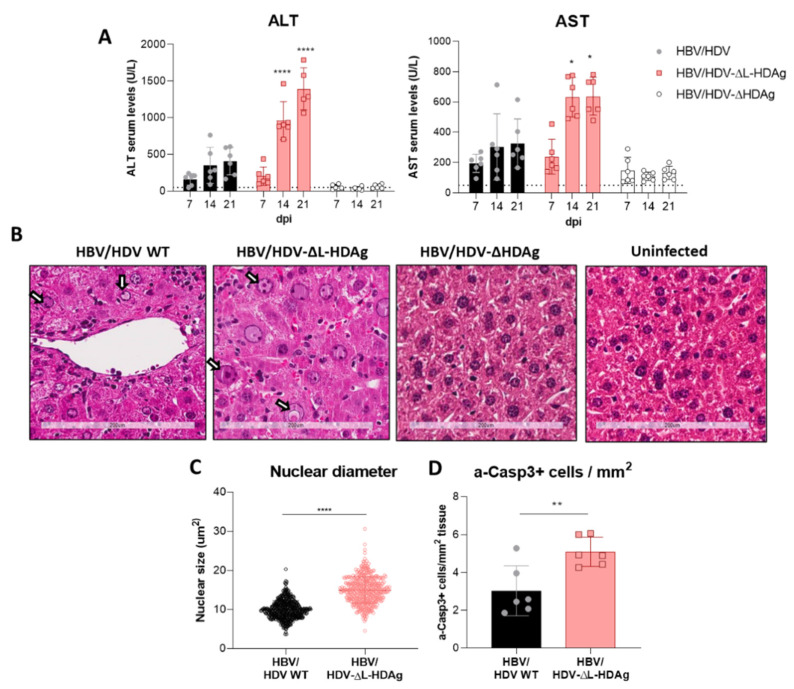
The expression of S-HDAg in the absence of L-HDAg resulted in increased liver damage. (**A**) Peripheral blood was collected every 7 days after vector injection to measure ALT and AST concentration in serum. Individual data points and mean values ± standard deviation are plotted; significant differences between groups at each time point are determined by a two-way ANOVA, followed by Bonferroni’s multiple-comparison test. (**B**) Liver sections from AAV-HBV/HDV WT-, AAV-HBV/HDV-ΔL-HDAg-, AAV-HBV/ HDV-ΔHDAg-injected mice that obtained 21 dpi were analyzed by H&E staining; an image of an uninfected liver was included as a control. Degenerated nuclei (white arrow) were observed in both groups of mice but were more abundant upon infection with the HDV-ΔL-HDAg mutant. Scale bar: 200 μm. (**C**) The nuclear diameter was significantly bigger when only S-HDAg was expressed. Significant differences were found by performing an unpaired *t*-test. (**D**) Immunostaining against cleaved caspase 3 revealed that S-HDAg overexpression exacerbates hepatocyte death by apoptosis. Statistical analysis using a Mann–Whitney test revealed differences between the two groups (* *p* < 0.05, ** *p* < 0.01, **** *p* < 0.0001).

**Figure 5 viruses-13-00788-f005:**
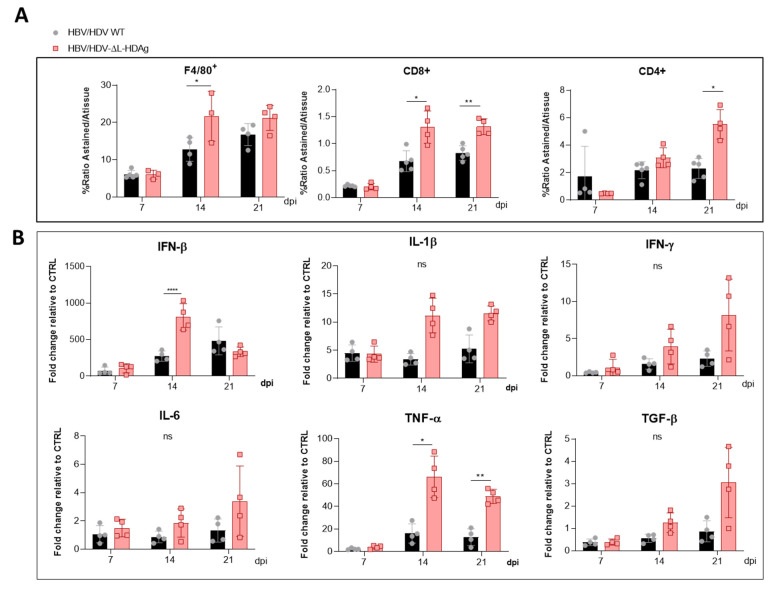
The HDV mutant expressing only S-HDAg increased the recruitment of inflammatory cells and the expression of cytokines. Liver sections collected at 7, 14 and 21 dpi of AAV-HBV/HDV WT- and AAV-HBV/HDV-ΔL-HDAg-injected mice were subjected to (**A**) F4/80-, CD4- and CD8 immunostaining to quantify the percentage of intrahepatic macrophages and CD4+ and CD8+ T lymphocytes, respectively, and (**B**) The expression levels of IFN-β, IFN-γ, IL-1β, IL-6, TNF-α and TGF-β were analyzed by RT-qPCR. Significant differences between groups at each time point were determined by a two-way ANOVA followed by Bonferroni’s multiple-comparison test. ns: no significant, * *p* < 0.05, ** *p* < 0.01, **** *p* < 0.0001.

**Figure 6 viruses-13-00788-f006:**
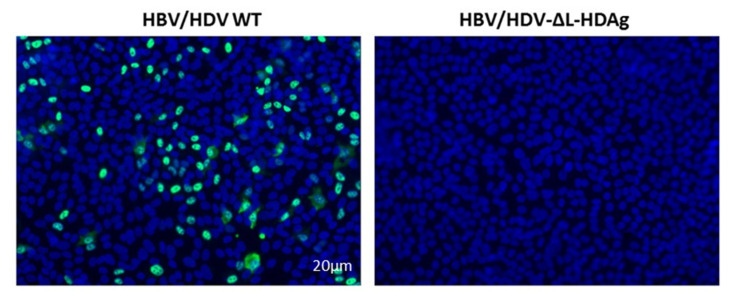
Infection of RagB6 mice with AAV-HBV/HDV-ΔL-HDAg failed to produce HDV infectious particles. Huh-7-hNTCP cells were incubated with serum collected from AAV-HBV/HDV WT- and AAV-HBV/HDV-ΔL-HDAg-infected RagB6 mice at 21-dpi. At 7-dpi, cells were fixed and immunostained to detect intracellular HDAg.

**Figure 7 viruses-13-00788-f007:**
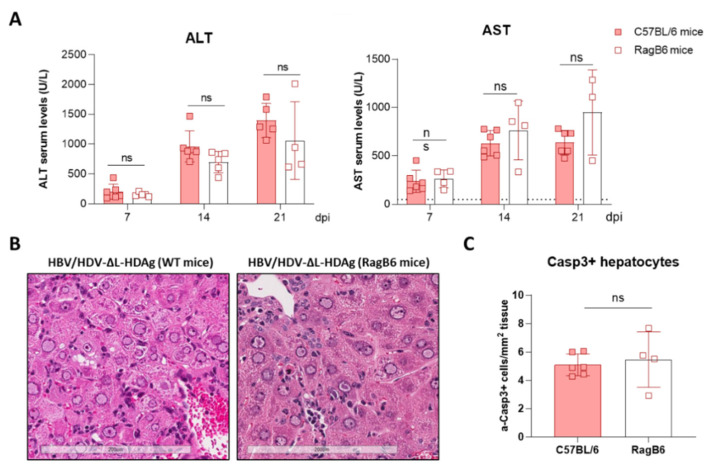
The liver damage triggered by the HDV mutant expressing only S-HDAg is independent of B and T lymphocytes and Natural Killer T (NKT) cells. Eight-week-old WT (*n* = 4) and RagB6 mice (*n* = 4) received 5 × 10^10^ vg of AAV-HDV-ΔL-HDAg and 5 × 10^10^ vg AAV-HBV. (**A**) Transaminase levels were analyzed at 7-, 14- and 21-dpi. Individual data points and mean values ± standard deviation are plotted; no significant differences were observed between WT and RagB6 mice injected with AAV-HBV/HDV-ΔL-HDAg by performing a two-way ANOVA and Bonferroni multiple-comparison test. (**B**) Examination of liver sections by H&E showed no histological differences between WT and RagB6 mice at 21 dpi. Scale bar: 200 μm. (**C**) Liver sections of WT and RagB6 mice receiving AAV-HDV-ΔL-HDAg sacrificed at 21 dpi were analyzed by IHC for activated Caspase 3. Mann–Whitney test did not reveal significant differences. ns: no significant.

**Figure 8 viruses-13-00788-f008:**
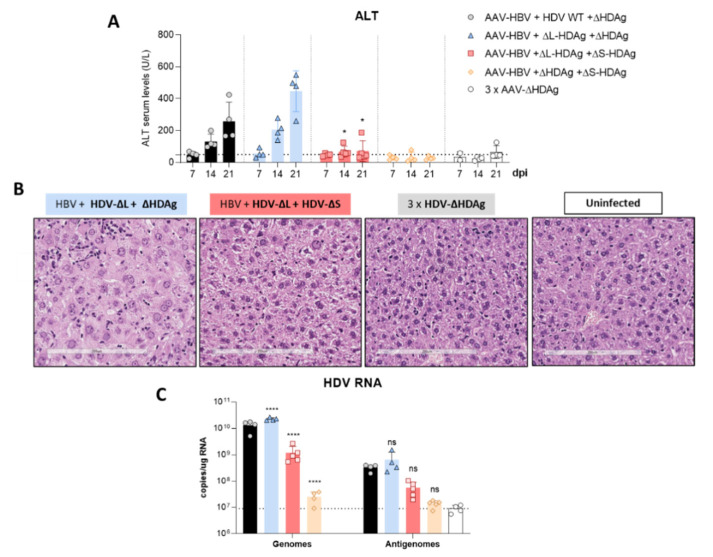
Expression of L-HDAg at the beginning of the HDV life cycle decreased liver damage and HDV replication. (**A**) Peripheral blood was collected weekly after the administration of the different combinations of AAV vectors, and ALT concentration in serum was measured. Individual data points and mean values ± standard deviation are plotted; significant differences between HBV/ΔL-HDAg/ΔHDAg and HBV/ΔL-HDAg/ΔS-HDAg groups at each time point were determined by a two-way ANOVA followed by Bonferroni multiple-comparison test. The dotted line represents the ALT ULN, 50 U/L. (**B**) Histological analysis of liver sections by H&E showed visible alterations in HBV/ΔL-HDAg/ΔHDAg that were not present in the HBV/ΔL-HDAg/ΔS-HDAg group. Normal histology was observed in the 3 × ΔHDAg and uninfected groups. Scale bar: 200 μm. (**C**) Total RNA was extracted from infected livers to quantify HDV genome and antigenome levels at 21 dpi by RT-qPCR. Significant differences between HBV/HDV WT/ΔHDAg and the rest of the groups were determined by a two-way ANOVA followed by Bonferroni multiple-comparison test. The dotted line represents the background. ns: no significant, * *p* < 0.05, **** *p* < 0.0001.

**Table 1 viruses-13-00788-t001:** Description of the antibodies used in immunohistochemistry (IHC).

Antibody	Produced in	Supplier	Cat#	Dilution
Anti-HDAg	Human	BioBank UNAV	-	1:10,000
Anti-CD45	Rat	BioLegend	103101	1:2000
Anti-CD4	Rabbit	Abcam	Ab183685	1:1000
Anti-CD8	Rabbit	Cell Signaling	98941	1:400
Anti-F4/80	Rat	BioLegend	123102	1:40,000
Anti-cleaved-Casp3	Rabbit	Cell Signaling	9661	1:200
Anti-human	Rabbit	Dako	P0214	1:3000
Anti-rabbit	Goat	Cell Signaling	7074S	1:5000
Anti-rat	Rabbit	Vector	BA4001	1:200

## Data Availability

Data is contained within the article or Appendix A.

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
