# Peer review of "AAV-HDV: An Attractive Platform for the In Vivo Study of HDV Biology and the Mechanism of Disease Pathogenesis†"

_viruses, 2021, doi:10.3390/v13050788_

Round 1
Reviewer 1 Report
The authors propose an interesting combination of AAV-HDV vector and mouse model to study in vivo HDV/Hepatocytes interaction allowing to explore HDV pathogenesis mechanism. They focused on the action of the small (S-HDAg) and the large (L-HDAg) delta protein on the viral replication and also on the damage in mouse hepatocytes. Apart from the model used, this research has already been studied especially in vitro [Abeywickrama-Samarakoon N et al. Nat Commun. 2020 Jan / Yamaguchi Y. et al Science. 2001 Jul / Xia YP et al J Virol. 1992 Nov / Chao M et al J Virol. 1990 Oct] but their results confirm in vivo the role of S-HDAg and L-HDAg on HDV replication and their localization in different cellular compartments. However, the low level of L-HD inflammatory action on the cell presented here seems to correspond to a different result from all previous shown for example in the following article [Park C-Y et al. Mol Cells. 2009 Jul]. Anyway, in vivo impact of each of the 2 proteins on hepatic inflammation demonstrated in this study remains an interesting element to explain the potentially fulminant nature of HBV / HDV infection in humans. In vitro and in vivo experiments are clearly exposed and the results are well integrated in their discussion to shed light and improve knowledge on the action of these 2 proteins in liver infection.
Major comments
- L-HDAg synthesis and its accumulation into several cell compartments have been shown to interfere with multiple cellular signaling pathways and induce a pathogenesis during infection [Wei Y et al. J Virol. 1998 / Park C-Y et al. Mol Cells. 2009 Jul / Williams V et al. J Viral Hepat. 2012 Oct]. Indeed, the overexpression of L-HDAg induces the transactivation a variety of heterologous promoters and upstream regulatory elements, notably by activating serum response factor-associated transcription. Park et al. showed in 2009 that L-HDAg is able to increase TNF-alpha-induced NF-kappa B transcriptional activation involved in inflammation pathways. I think the authors should integrate this information into their discussion and explain why their results indicate that S-HDAg is mainly responsible of inflammatory cells? Thus, regarding results shown in figure 4 and 5, did the authors have made the same experiments with AAV-HBV/HDV-∆S-HDVAg in order to see the effect of L-HDAg alone, especially on ALT, AST, TNF-α and TGF-β levels and nuclear diameter?
- How the authors explain the absence of HDV genome in HepG2 cells at day 1 and 3 whereas they detect HDV antigenome? (Figure 1B and page 5 line 213).
- In the same way, in figure 2B and figure 3A authors do not detect HDV genome for HDV-∆S-HDAg and HDV-∆HDAg in vitro (Huh7 cells) and in vivo (liver of infected mice) whereas antigenomes were detected? It’s known that L-HDAg inhibits HDV replication but does this mean that L-HDAg interferes preferentially in genome synthesis from antigenome during double rolling cycle of HDV replication? This important and little-known point is not mentioned in the discussion.
Minor comments
- Page7 line 316: The authors describe 5 groups of experiment but the indication “(n=4)” is not clear. What does this number correspond to?
- Page15 line 467: I think it will be better to replace “found” by “confirmed” as this role of L-HDAg was clearly previously described [Lee CH, Chang SC, Wu CH, Chang MF. A novel chromosome region maintenance 1-independent nuclear export signal of the large form of hepatitis delta antigen that is required for the viral assembly. J Biol Chem. 2001 Mar].
- The authors could reference and integrate in the discussion the article by Cole SM et al of 1993 [Cole SM, Macnaughton TB, Gowans EJ. Differential roles for HDAg-p24 and -p27 in HDV pathogenesis. Prog Clin Biol Res. 1993;382:131–8.], which evokes the different role of the 2 delta proteins in HDV pathogenesis.
Reviewer 2 Report
This manuscript exploits a hepatitis delta virus system (previously developed by the same lab) to examine the contribution of the two viral antigens to replication, assembly and pathogenesis. While somewhat incremental, the work does function as a proof of concept for the use of the system to perform molecular virology experiments in vivo. In this case the focus is on demonstrating the distinct roles of the small and large antigens. The manuscript is well written for the most part and clearly describes the work. Specific comments: Many abbreviations are not defined at first use. Line 44 - a brief statement of the current standard of care for HBV/HDV would be nice here. Line 51 - “defective” doesn’t really fit. Satellite virus, as used previously, seems more suitable. Line 151-2 - 1% PMSF suggests that this was originally prepared as a 100x stock solution. It would be more helpful to describe the final concentration. Similar for the other two inhibitors used. Also, what protease inhibitor cocktail was used? Line 163 - Tween-20? Line 174 - Reverse transcribed? Line 220 - “support a continue replication” is unclear Fig. 1D - The spike of IFN signalling at day 10 is a little odd, as it does not appear correlate with a specific behaviour of the virus. Why is there no signalling at day 7 or 14? This feels like a silly question, but can the authors be certain that this spike in signalling was not related to the splitting of the cells? Can any explanation be provided for this apparently transient signal appearing at 10 days then disappearing again? Line 292-3 - Perhaps I have misunderstood, but isn't TGFb an anti-inflammatory cytokine? On that theme, would we expect to see reduced anti-inflammatory signals, or does increased TGFb represent an attempt by the liver to maintain homeostasis? In any case, this section would benefit from some additional explanation. Fig. 2A - A minor thing, but these blots could be lined up to facilitate comparison of the molecular weights of the bands. Fig. 2D - it would be good to show the accompanying western blots, including some verification of the successful fractionation, i.e. blots for markers of the three compartments. Fig. 4B and 7B - an uninfected micrograph form comparison would be helpful. Fig. 7C - should have a key for the colours. Biological replicates? It is generally unclear how many independent replicates have been performed. For the in vivo experiments it is sometimes explicit how many mice were used, but not in every legend. For the in vitro experiments it is unclear whether the data are from independent biological replicates or technical replicates from the same experiment.
Reviewer 3 Report
Maestro et al. present interesting new information regarding connections between expression of the two forms of the hepatitis delta antigen and pathogenesis, using an AAV vector delivery system to infect hepatocytes in mice with mutated forms of the HDV genome. Although interesting, the authors do not do enough to consider or discuss how their AAV system differs from other methods to look at HDV pathogenesis. Also, some of the discussion of the effects of varying amounts of S-HDAg and L-HDAg is too simplistic and misses a number of important publications.
With the AAV-vectored HDV system, the authors of this study are able to see clear evidence of liver damage – highly elevated levels of ALT and AST, degenerated nuclei and even cleaved caspase 3 as a sign of apoptosis – whereas numerous other studies in mice and most studies in woodchucks have not seen such disease. Why is that? One possibility is that by using massive doses of the AAV-vectored HDV (5 x 10^5 genomes), the authors are able to achieve simultaneous infection of what looks like close to 25% of the hepatocytes in the liver. In this case, the authors have a powerful system with unique abilities to assess HDV pathogenesis. Another possibility is that the AAV-vectored system somehow boosts HDV replication to levels higher than would otherwise occur in cells, perhaps due to the use of the powerful alpha-1 anti-trypsin promoter. (One can see the effects of this promoter in Figure 2A, in which the HDV antigenome and L-HDAg are detectable even without HDV replication.) This second scenario is less interesting because the relevance to HDV infection would be less clear. The authors should attempt to rule out the second possibility by comparing HDV replication following infection of cultured cells with their AAV system on the one hand, and authentic HDV on the other. An alternative would be to use AAV-vectored HDV without a strong promoter – cells transfected with over-length HDV cDNA constructs initiate replication quite well.
An advantage of the AAV delivery system is that the consequences of the mutations analyzed here (one which abrogates only S-HDAg expression, the other only L-HDAg expression) could not be easily assessed using infectious HDV particles - the authors make this point in passing in the Discussion, but should make this particular point more clearly earlier in the paper – e.g. in the Results section.
Additional points:
Introduction
- line 50 – although there has been some evidence that HDAg can be directly cytotoxic, the question is largely unsettled. The 293HDAg cells created by the Taylor group did not show any evidence of cytotoxicity.
- Editing during replication was shown by two groups to be due to just one of the ADARs – ADAR-1
Results
- Line 228 – 230 (and 454 – 455). There is certainly a negative correlation between levels of HDV replication in the two cell lines and their abilities to mount innate responses, but the correlation does not allow one to conclude that the variations in innate responsiveness are responsible for the different replication levels.
- Figures 1 and2. The results presented in Figures 1 and 2 provide little in the way of new information because they mostly replicate results published more than 20 years ago by several labs. The data is useful as supporting information to describe the AAV delivery system.
- Figure 2 – When comparing the amounts of genome RNA and HDAg in Figs. 1 and 2, it appears that the amount of replication in cells transfected with the HDV-∆L-HDAg construct is substantially lower than for the wt. What is responsible for this variation?
- Figure 2 – the cell fractionation analysis should include standards for known cytosolic, nuclear and membrane proteins.
- Figure 7 – To fully make the conclusion that adaptive immune responses are not responsible for liver damage in animals infected with the AAV-HDV-∆L-HDag, the authors should also include the wt HDV construct in the RagB mice.
Discussion
- Line 458 correct Zang to Zhang
- Line 471 – 473. The statement is incorrect. Amounts of HDAg were about 2.5-fold higher in animals infected with the ∆L HDAg virus (Fig. 3C) but replication, as measured by RNA levels, was not statistically different from the wt (Fig. 3A).
- Lines 488 – 492. The papers cited do not support the statements made here.
- Lines 497 – 506. This paragraph is confusing and does not consider the issue in sufficient depth. There are three sources of L-HDAg – viruses encoding L-HDAg in the inocula that patients “receive”, viruses encoding L-HDAg produced during the course of the infection, and L-HDAg produced within infected cells as a result of RNA editing. Infectious HDV inocula will certainly contain a mixture of virus particles – some with genomes encoding S-HDAg, others with genomes encoding L-HDAg. However, because most individuals will be exposed to very small numbers of genomes relative to the number of hepatocytes in the liver, viruses encoding L-HDAg in the inoculum are unlikely to have any meaningful impact on the infection. However, as infection progresses, particularly in individuals with chronic HBV (and who are thus unlikely to clear HDV) it is interesting to consider what might happen as levels of virus increase. Virus particles harboring genomes encoding L-HDAg will enter many cells but will be unable to replicate unless those cells are also infected with particles encoding S-HDAg; even then, replication is likely to be strongly suppressed by L-HDAg produced from the genomes encoding L-HDAg. The infection of hepatocytes by L-HDAg-encoding viruses could thus limit the progression of virus spread through the liver. Finally, L-HDAg produced within infected cells as a result of editing occurring during replication in those cells could limit replication and perhaps reduce cytotoxic effects due to high replication levels. Chao et al. suggested this in 1990.
Round 2
Reviewer 3 Report
With the included changes the manuscript is improved and acceptable.
Author Response
Thank you very much for your support